# Dostarlimab or pembrolizumab plus chemotherapy in previously untreated metastatic non-squamous non-small cell lung cancer: the randomized PERLA phase II trial

PERLA is a global, double-blind, parallel phase II trial (NCT04581824) comparing efficacy and safety of anti–PD-1 antibodies dostarlimab and pembrolizumab, plus chemotherapy (DCT and PCT, respectively) as first-line treatment in patients with metastatic non-squamous NSCLC without known targetable genomic aberrations. Patients stratified by PD-L1 tumor proportion score and smoking status were randomized 1:1, receiving ≤35 cycles 500 mg dostarlimab or 200 mg pembrolizumab, ≤35 cycles 500 mg/m² pemetrexed and ≤4 cycles cisplatin (75 mg/m²) or carboplatin (AUC 5 mg/ml/min) Q3W. Primary endpoint was overall response rate (ORR) (blinded independent central review). Secondary endpoints include progression-free survival (PFS) based on investigator assessment, overall survival (OS) and safety. Exploratory endpoints include ORR by PD-L1 subgroup and duration of response. PERLA met its pre-specified endpoint. ORR (n/N; 95% CI) is 45% (55/121; 36.4–54.8) for DCT and 39% (48/122; 30.6–48.6) for PCT (data cut-off: 07 July 23), numerically favoring dostarlimab in PD-L1-positive subgroups. Median PFS (months [95% CI]) is 8.8 (6.7–10.4) for DCT and 6.7 (4.9–7.1) for PCT (HR 0.70 [95% CI: 0.50–0.98]; data cut-off: 04 August 22). Median OS (months [95% CI]) is 19.4 (14.5–NR) for DCT and 15.9 (11.6–19.3) for PCT (HR 0.75 [95% CI: 0.53–1.05]) (data cut-off: 07 July 23). Safety profiles are similar between groups. In this study, DCT shows similar efficacy to PCT and demonstrates clinical efficacy as first-line treatment for patients with metastatic non-squamous NSCLC.

Lung cancer is one of the most common forms of cancer globally, with an estimated 1.8 million related deaths in 2020 alone[1]. Non-small cell lung cancer (NSCLC) accounts for 80–90% of lung cancers[2,3]. In the United States (US), over half of patients with NSCLC have advanced or metastatic disease at diagnosis. Until recently this was associated with very poor prognosis despite available treatments; the 5-year survival rate is only 7% for metastatic disease[4]. As such, there was a clear need for novel therapeutic agents for NSCLC, especially for first-line (1 L) treatment[5].

In recent years, outcomes for patients with NSCLC in the 1 L setting have improved with the introduction of targeted therapies for patients with oncogenic drivers, and immunotherapies for those

✉e-mail: LIMLOVE2008@yuhs.ac

without targetable oncogenic drivers. Immunotherapies include programmed death receptor-1/programmed death-ligand 1 (PD-[L]1) inhibitors, either as monotherapy (for patients with high levels of PD-L1 expression) or in combination with chemotherapy (irrespective of PD-L1 expression)[5–7]. Several PD-(L)1 inhibitors have been approved or are in development for the treatment of NSCLC, including pembrolizumab, dostarlimab, nivolumab, and atezolizumab, and have shown promising efficacy and tolerable safety profiles in patients with advanced/metastatic NSCLC[5–8].

Dostarlimab is an anti-PD-1 monoclonal antibody approved as a monotherapy in the European Union for recurrent or advanced mismatch repair-deficient (dMMR)/microsatellite instability-high (MSI-H) endometrial cancer that has progressed on or after platinum-based chemotherapy and in the US for recurrent or advanced dMMR endometrial cancer that has progressed on or after platinum-based chemotherapy as well as dMMR solid tumors following prior treatment and with no alternative treatment options[8–10]. Dostarlimab is also approved in the US in combination with carboplatin and paclitaxel followed by dostarlimab as a single agent for primary recurrent or advanced dMMR/MSI-H endometrial cancer[10]. In the recent first-in-human, multicenter, Phase I, open-label GARNET trial (NCT02715284), dostarlimab monotherapy showed promising antitumor activity in recurrent or advanced NSCLC across various PD-L1 expression subgroups and an acceptable safety profile, with an overall response rate (ORR) of 26.9% ($n = 18/67$)[11]. Dostarlimab has also demonstrated significantly improved outcomes in a small subgroup ($n = 12$) of patients with dMMR locally advanced rectal cancer, with a complete clinical response observed in all 12 patients at 6 months of follow-up[12]. Considering these positive results and a lack of head-to-head clinical trials of anti-PD-1 treatments in metastatic NSCLC, further studies of dostarlimab in combination with chemotherapy are warranted to evaluate the efficacy of dostarlimab as a PD-1 inhibitor backbone for combination therapy and to determine optimal PD-1 inhibitor therapies for different NSCLC patient subgroups.

Here, we show that in the PERLA study, dostarlimab plus chemotherapy has comparable efficacy to pembrolizumab plus chemotherapy as 1 L treatment for patients with metastatic non-squamous NSCLC without targetable oncogenic driver mutations. Safety profiles are also comparable and consistent with published data in PD-(L)1 inhibitors.

## Results

Two data cuts have been used for the analyses in this manuscript. PERLA primary analyses were conducted using data available as of August 4, 2022. Due to the immature overall survival (OS) and duration of response (DOR) data from the primary analyses, additional analyses were planned for more mature data and better understanding of these endpoints; pre-planned updated analyses were conducted using data available as of July 7, 2023. To provide data on the primary outcome (ORR) and safety using the longest available follow-up, ORR and safety analyses from the July 7, 2023 data cut are included in the main manuscript; results from the August 4, 2022 data cut are available in the Supplementary Information. PFS was a secondary endpoint, and results from the August 4 data cut are presented here in the main text; based on the maturity of data at the time of the primary analysis, PFS was not re-analyzed in the updated analysis. OS and DOR analyses using the July 7, 2023 data cut are reported in the main manuscript.

### Study population and baseline characteristics

The PERLA study design is shown in Fig. 1 (see "Methods" section for full details). A total of 243 patients were enrolled in the study and randomized to either dostarlimab plus chemotherapy or pembrolizumab plus chemotherapy (Fig. 2). The first patient was enrolled on November 19, 2020, and the last patient on February 18, 2022. The trial is active but no longer recruiting as the predetermined sample size has been reached. Time from first subject randomized to August 4, 2022, data cut-off was 20 months and the time from last subject randomized to August 4, 2022, data cut-off was 5 months. For OS analyses, time from first subject randomized to July 7, 2023, data cut-off was 31 months and the time from last subject randomized to July 7, 2023, data cut-off was 16 months. Baseline demographics and disease characteristics were broadly similar between treatment groups (Table 1). There were slightly fewer patients aged ≥65 years and fewer female patients in the dostarlimab plus chemotherapy group than in the pembrolizumab plus chemotherapy group (46% vs 53% and 30% vs 37%, respectively). In addition, there was a greater percentage of patients with an Eastern Cooperative Oncology Group (ECOG) performance status of 1 in the dostarlimab plus chemotherapy group than the pembrolizumab plus chemotherapy group (69% vs 59%, respectively), as well as a greater proportion of patients with brain and liver metastases (18% vs 12% and 16% vs 11%, respectively) (Table 1). Overall, 41% and 42% of

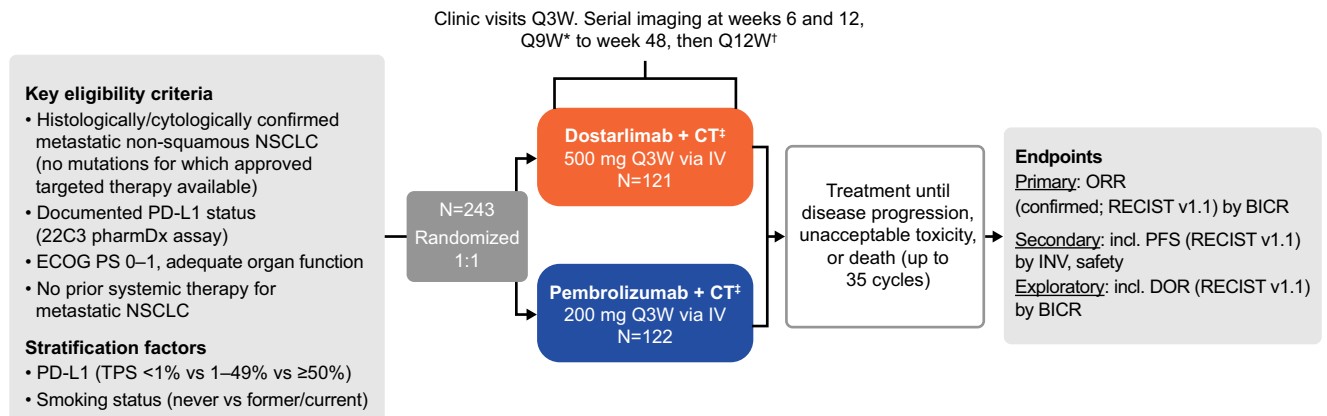

**Fig. 1 | Study design.** *Or more frequently, if clinically indicated. †Imaging will continue to be performed until discontinuation of study treatment due to disease progression with clinical instability, start of subsequent anticancer treatment, withdrawal of informed consent, or death, whichever comes first. ‡Up to 35 cycles pemetrexed with either cisplatin or carboplatin for the first four cycles. BICR blinded independent central review, CT chemotherapy, DOR duration of response, ECOG PS Eastern Cooperative Oncology Group performance status, INV investigator assessment, IV intravenous, NSCLC non-small cell lung cancer, ORR overall response rate, PD-L1 programmed death ligand 1, PFS progression-free survival, QxW every x weeks, RECIST v1.1 Response Evaluation Criteria in Solid Tumors version 1.1, TPS tumor proportion score.

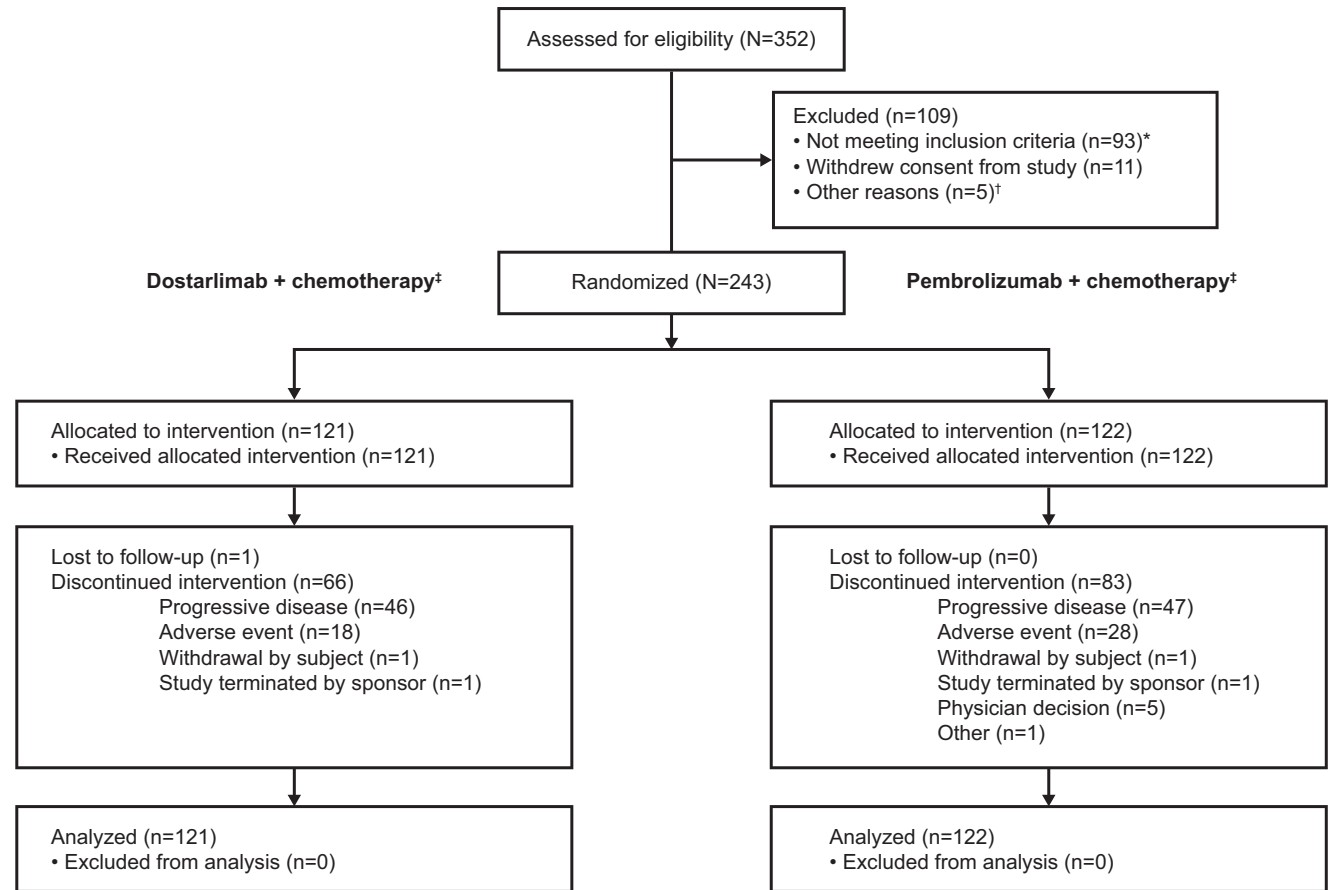

**Fig. 2 | CONSORT flow diagram.** *Most common reasons were patients not meeting the inclusion criteria for confirmed metastatic non-squamous NSCLC with documented absence of genomic aberration with available approved target therapy (*n* = 24) and for documented PD-L1 status by the 22C3 pharmDx assay (*n* = 22).

†Other reasons included death (*n* = 4) and lost to follow-up (*n* = 1). ‡Pemetrexed plus platinum-based therapy (either cisplatin or carboplatin). CONSORT Consolidated Standards of Reporting Trials, NSCLC non-small cell lung cancer, PD-L1 programmed death ligand 1.

patients had PD-L1 tumor proportion score (TPS) < 1% for dostarlimab plus chemotherapy and pembrolizumab plus chemotherapy, respectively. The proportion of patients from each enrollment region was similar between treatment groups, with most patients enrolled from Europe and South America (Table 1). Most patients in both treatment arms received carboplatin (dostarlimab plus chemotherapy: 99/121 [82%]; pembrolizumab plus chemotherapy: 108/122 [89%]) as opposed to cisplatin (dostarlimab plus chemotherapy: 22/121 [18%]; pembrolizumab plus chemotherapy: 14/122 [11%]).

**Overall response rate**

At data cut-off of July 7, 2023, patients in the dostarlimab plus chemotherapy group had received a mean of 15.0 cycles of dostarlimab (standard deviation: 10.9) with a median (range) duration of exposure of 8.97 months (0.3–26.8). Patients in the pembrolizumab plus chemotherapy group had received a mean of 11.3 cycles of pembrolizumab (SD: 10.4) with a median duration of exposure (range) of 5.63 months (0.2–25.3). For dostarlimab plus chemotherapy, the confirmed ORR per Response Evaluation Criteria in Solid Tumors (RECIST) version (v) 1.1 based on blinded independent central review (BICR) was 45% (55/121; 95% confidence interval [CI]: 36–55%), with four complete responses (CRs) (3%) and 51 partial responses (PRs) (42%). For pembrolizumab plus chemotherapy, ORR was 39% (48/122; 95% CI: 31–49%), with six CRs (5%) and 42 PRs (34%) (6% difference [80% CI: −1.95–14.02%; 95% CI: −6.17–18.24%]) (Table 2). Five patients (4%) in the dostarlimab treatment group and 13 patients (11%) in the pembrolizumab treatment group had unknown or missing responses and were classified as 'not done' (Table 2; Supplementary Table 4). These

results are supported by the analysis of ORR based on investigator assessment, which was 43% (52/121; 95% CI: 34–52%) in the dostarlimab plus chemotherapy treatment group and 30% (37/122; 95% CI: 22–39%) in the pembrolizumab plus chemotherapy group.

Subgroup analyses of ORR based on BICR by PD-L1 status are shown in Table 2. The highest ORR was observed in the TPS ≥ 50% subgroups of both dostarlimab plus chemotherapy (74% [20/27; 95% CI: 54–89%]) and pembrolizumab plus chemotherapy treatment groups (48% [13/27; 95% CI: 29–68%]). Patients in the PD-L1-positive (TPS ≥ 1%) subgroup treated with dostarlimab plus chemotherapy had an ORR of 59% (42/71; 95% CI: 47–71%), while those treated with pembrolizumab plus chemotherapy had an ORR of 39% (28/71; 95% CI: 28–52%). ORR for PD-L1-negative (TPS < 1%) patients was 26% (13/50; 95% CI: 15–40%) in the dostarlimab plus chemotherapy subgroup and 39% (20/51; 95% CI: 26–54%) in the pembrolizumab plus chemotherapy subgroup.

Median DOR was 12.4 months (95% CI: 8.3–17.9) for patients treated with dostarlimab plus chemotherapy and 14.4 months (95% CI: 9.7–not reached) for those treated with pembrolizumab plus chemotherapy (Supplementary Fig. 1) at 54% maturity (56 events in a total of 103 patients).

Primary analyses of ORR at a data cut-off of August 4, 2022, are described in detail in Supplementary Information and shown in Supplementary Table 5. BICR-assessed maximum percentage reduction from baseline in tumor measurement is shown in Supplementary Fig. 2. Subgroup analyses of ORR based on BICR by PD-L1 status are also shown in Supplementary Table 6 and the differences in ORR between dostarlimab plus chemotherapy and pembrolizumab plus

## Table 1 | Demographics and baseline characteristics for patients receiving dostarlimab + chemotherapy and pembrolizumab + chemotherapy (ITT population)

| Variable | Dostarlimab + chemotherapy (N = 121) | Pembrolizumab + chemotherapy (N = 122) |
|---|---|---|
| **Median age, years (range)** | 64 (25–80) | 65 (46–86) |
| **Age group (years), *n* (%)** | | |
| <65 | 65 (54) | 57 (47) |
| ≥65 | 56 (46) | 65 (53) |
| **Sex, *n* (%)** | | |
| Male | 85 (70) | 77 (63) |
| Female | 36 (30) | 45 (37) |
| **Ethnicity, *n* (%)** | | |
| Hispanic or Latino | 25 (21) | 32 (26) |
| Other | 90 (74) | 84 (69) |
| Not reported[a] | 3 (2) | 5 (4) |
| Unknown[b] | 3 (2) | 1 (<1) |
| **Race, *n* (%)** | | |
| White | 87 (72) | 84 (69) |
| Asian | 23 (19) | 21 (17) |
| Unknown | 4 (3) | 6 (5) |
| Multiple | 3 (2) | 3 (2) |
| Black or African American | 1 (<1) | 3 (2) |
| American Indian or Alaska Native | 1 (<1) | 0 (0) |
| Native Hawaiian or Other Pacific Islander | 0 (0) | 0 (0) |
| Not Reported | 2 (2) | 5 (4) |
| **Enrollment region[c], *n* (%)** | | |
| Europe | 62 (51) | 65 (53) |
| South America | 35 (29) | 34 (28) |
| East Asia | 23 (19) | 21 (17) |
| USA | 1 (<1) | 2 (2) |
| **Smoking status[d], *n* (%)** | | |
| Never smoked | 17 (14) | 17 (14) |
| Former/current smoker | 104 (86) | 105 (86) |
| **ECOG performance status, *n* (%)** | | |
| 0 | 37 (31) | 50 (41) |
| 1 | 84 (69) | 72 (59) |
| **Stage at initial diagnosis[e], *n* (%)** | | |
| I | 11 (9) | 9 (7) |
| II | 2 (2) | 3 (2) |
| III | 4 (3) | 9 (7) |
| IV | 101 (83) | 100 (82) |
| Unknown | 3 (2) | 1 (<1) |
| **Histology, *n* (%)** | | |
| Non-squamous | 117 (97) | 121 (>99) |
| Mixed | 4 (3)[f] | 1 (<1)[g] |
| **PD-L1 status[d], *n* (%)** | | |
| TPS < 1% | 50 (41) | 51 (42) |
| TPS 1–49% | 44 (36) | 44 (36) |
| TPS ≥ 50% | 27 (22) | 27 (22) |
| TPS ≥ 1% | 71 (59) | 71 (58) |

## Table 1 (continued) | Demographics and baseline characteristics for patients receiving dostarlimab + chemotherapy and pembrolizumab + chemotherapy (ITT population)

| Variable | Dostarlimab + chemotherapy (N = 121) | Pembrolizumab + chemotherapy (N = 122) |
|---|---|---|
| **Metastases at baseline, *n* (%)** | | |
| Bone | 39 (32) | 34 (28) |
| Brain | 22 (18) | 15 (12) |
| Liver | 19 (16) | 14 (11) |

*ECOG* Eastern Cooperative Oncology Group, *ITT* intention-to-treat, *NSCLC* non-small cell lung cancer, *PD-L1* programmed cell death ligand-1, *TPS* tumor proportion score.
[a]Not reported indicates cases where a patient prefers to not provide their ethnicity or where the collection of this data is not permitted according to local regulations.
[b]Unknown indicates cases where these data are not known.
[c]East Asia – Republic of Korea, Taiwan; Europe – France, Germany, Italy, Poland, Romania, Spain; South America – Argentina, Brazil, Chile.
[d]Randomization factors based on data collected in Interactive Response Technology at randomization.
[e]Patients are required to have metastatic NSCLC at enrollment.
[f]Predominantly non-squamous histology without small cell component (*n* = 2) and other (*n* = 2).
[g]Predominantly non-squamous histology without small cell component.

chemotherapy across prespecified subgroups are shown in Supplementary Fig. 3.

### Progression-free survival
As of August 4, 2022, the median follow-up time for PFS was 9.1 months (IQR: 6.8–11.4) for dostarlimab plus chemotherapy and 9.0 months (IQR: 6.7–11.2) for pembrolizumab plus chemotherapy. Median PFS was 8.8 months (95% CI: 6.7–10.4) for patients treated with dostarlimab plus chemotherapy and 6.7 months (95% CI: 4.9–7.1) for those treated with pembrolizumab plus chemotherapy (Table 3 and Fig. 3a) at 57% maturity (138 events in a total of 243 patients). The PFS hazard ratio (HR) was 0.70 (95% CI: 0.50–0.98) for dostarlimab plus chemotherapy versus pembrolizumab plus chemotherapy. The PFS rate at 9 months was 46% (95% CI: 36–56%) for patients treated with dostarlimab plus chemotherapy and 36% (95% CI: 26–45%) for those treated with pembrolizumab plus chemotherapy (Table 3). In subgroup analyses of PFS by PD-L1 status, the lowest HR (0.60; 95% CI: 0.27–1.29) occurred in the TPS ≥ 50% group (*n* = 27 per treatment group) (Supplementary Table 7).

### Overall survival
As of July 7, 2023, the median follow-up time for OS was 20.7 months (95% CI: 19.3–22.3) for dostarlimab plus chemotherapy and 21.6 months (95% CI: 19.6–23.1) for pembrolizumab plus chemotherapy. Median OS was 19.4 months (95% CI: 14.5–not reached) for patients treated with dostarlimab plus chemotherapy and 15.9 months (95% CI: 11.6–19.3) for those treated with pembrolizumab plus chemotherapy (Table 4 and Fig. 3b) at 55% maturity (134 events in a total of 243 patients). OS HR for dostarlimab plus chemotherapy versus pembrolizumab plus chemotherapy was 0.75 (95% CI: 0.53–1.05) (Fig. 3b). At 12 months, the OS rate was 63% (95% CI: 54–71%) for patients in the dostarlimab plus chemotherapy group and 58% (95% CI: 48–66%) for patients in the pembrolizumab plus chemotherapy group (Table 4). Kaplan–Meier curves for OS by PD-L1 TPS subgroups are provided in Supplementary Fig. 4.

Primary OS analyses are reported in Supplementary Table 8.

### Safety
Overall, the safety profiles of dostarlimab plus chemotherapy and pembrolizumab plus chemotherapy were similar (data cut-off: July 7, 2023) (Table 5; Fig. 4). The proportion of patients experiencing any treatment-emergent adverse event (TEAE) was the same for both

**Table 2 | ORRs for patients receiving dostarlimab + chemotherapy and pembrolizumab + chemotherapy (ITT population as of July 7, 2023)**

| | Dostarlimab + chemotherapy (N = 121) | Pembrolizumab + chemotherapy (N = 122) |
|---|---|---|
| **Best overall response, n (%)** | | |
| Complete response | 4 (3) | 6 (5) |
| Partial response | 51 (42) | 42 (34) |
| Stable disease | 49 (40) | 48 (39) |
| Progressive disease | 12 (10) | 12 (10) |
| Not evaluable | 0 | 1 (<1) |
| Not done[a] | 5 (4) | 13 (11) |
| **ORR, n (%)** | | |
| Complete response + partial response 95% CI | 55 (45) 36.4–54.8 | 48 (39) 30.6–48.6 |
| **ORR by PD-L1 TPS subgroup, n/N (%)** | | |
| TPS < 1% 95% CI | 13/50 (26) 15–40 | 20/51 (39) 26–54 |
| TPS ≥ 1% 95% CI | 42/71 (59) 47–71 | 28/71 (39) 28–52 |
| TPS 1–49% 95% CI | 22/44 (50) 35–65 | 15/44 (34) 21–50 |
| TPS ≥ 50% 95% CI | 20/27 (74) 54–89 | 13/27 (48) 29–68 |

ORR was assessed per RECIST v1.1 based on BICR.

*BICR* blinded independent central review, *CI* confidence interval, *ITT* intention-to-treat, *ORR* overall response rate, *PD-L1* programmed death ligand-1, *RECIST v1.1* Response Evaluation Criteria in Solid Tumors version 1.1, *TPS* tumor proportion score.

[a]Patients with ORR listed as 'not done' had unknown or missing responses. The reasons for unknown or missing responses in each arm are listed in Supplementary Table 4.

**Table 3 | PFS for patients receiving dostarlimab + chemotherapy and pembrolizumab + chemotherapy (ITT population as of August 4, 2022)**

| Variable | Dostarlimab + chemotherapy (N = 121) | Pembrolizumab + chemotherapy (N = 122) |
|---|---|---|
| Median PFS follow-up time, months (IQR) | 9.1 (6.8–11.4) | 9.0 (6.7–11.2) |
| PFS events observed, n | 64 | 74 |
| Median PFS (95% CI), months | 8.8 (6.7–10.4) | 6.7 (4.9–7.1) |
| **Estimated probability of PFS, % (95% CI)** | | |
| 6 months | 61 (52–70) | 52 (42–61) |
| 9 months | 46 (36–56) | 36 (26–45) |
| Hazard ratio (95% CI) | 0.70 (0.50–0.98) | |

PFS was assessed per RECIST v1.1 based on investigator assessment.

*CI* confidence interval, *IQR* interquartile range, *ITT* intention-to-treat, *PFS* progression-free survival, *RECIST v1.1* Response Evaluation Criteria in Solid Tumors version 1.1.

treatment groups (98%). The most frequent TEAEs were largely balanced between the two groups (Supplementary Table 9). The proportion of patients experiencing any treatment-related adverse event (TRAE) was similar between groups (85% for dostarlimab plus chemotherapy and 81% for pembrolizumab plus chemotherapy). The four most frequent TRAEs (related to any study treatment) in both the dostarlimab plus chemotherapy and pembrolizumab plus chemotherapy groups were anemia (41% and 39%, respectively), diarrhea (35% and 38%, respectively), asthenia (21% and 23%, respectively), nausea (both 20%), and neutropenia (15% and 18%, respectively) (Supplementary Table 9). The most frequent TRAE related to dostarlimab or pembrolizumab specifically was anemia (12%) in the dostarlimab plus chemotherapy group and asthenia (14%) in the pembrolizumab plus chemotherapy group (Supplementary Table 9).

While more patients experienced dostarlimab-related AEs (71%) than pembrolizumab-related AEs (57%), a numerical trend favoring dostarlimab was observed in the proportion of patients experiencing immune-related adverse events (irAEs) (31% for dostarlimab plus chemotherapy and 39% for pembrolizumab plus chemotherapy), serious adverse events (SAEs) (41% and 48%, respectively), AEs leading to treatment discontinuation (29% and 38%, respectively), and AEs leading to immunotherapy discontinuation (17% and 24%, respectively) (Table 5). Fatal TRAEs were observed in 2% of patients in the dostarlimab plus chemotherapy group and 4% of patients in the pembrolizumab plus chemotherapy group; individual fatal TRAEs are also summarized in Table 5. Primary safety analyses are described in detail in Supplementary Information and are shown in Supplementary Tables 10 and 11.

## Discussion

This Phase II, randomized, double-blind trial evaluated the efficacy of dostarlimab plus chemotherapy and pembrolizumab plus chemotherapy in newly diagnosed patients with metastatic, non-

squamous NSCLC without known targetable oncogenic driver mutations, with the primary endpoint (ORR) assessed based on BICR. An earlier small (N = 68) single-center randomized trial compared the PD-1 inhibitors sintilimab and pembrolizumab as 1 L treatment in advanced NSCLC; however, to our knowledge, the PERLA trial is the first global study that prospectively compared anti-PD-1 therapies head-to-head[13,14]. Dostarlimab has previously demonstrated promising efficacy and a manageable safety profile as a monotherapy in patients with recurrent or advanced post-platinum-based chemotherapy NSCLC in the GARNET trial[11]. Our findings from the PERLA trial indicate that efficacy and safety were generally comparable between patients treated with dostarlimab plus chemotherapy and those treated with pembrolizumab plus chemotherapy, with a numerical trend favoring dostarlimab in several parameters (i.e., ORR, PFS and OS data).

ORR was similar across both treatment groups and at both datacuts, though numerical differences in ORR were observed overall and between subgroups, with a 6% and 13% difference favoring dostarlimab, by BICR and investigator-assessment, respectively, in the updated analysis. ORR based on BICR numerically favored dostarlimab plus chemotherapy in the PD-L1-positive subgroups (TPS ≥ 1%, TPS 1–49%, and TPS ≥ 50%); however, this study was not formally designed to demonstrate, or statistically powered to test for, superiority. The ORR for dostarlimab plus chemotherapy in this analysis was similar to that observed for pembrolizumab plus chemotherapy in the KEYNOTE-189 trial (45% and 48%, respectively), which evaluated pembrolizumab plus the same chemotherapy regimen (pemetrexed plus cisplatin or carboplatin) versus placebo plus chemotherapy in patients with metastatic non-squamous NSCLC[15]. The pembrolizumab plus chemotherapy arm in PERLA had a lower ORR than the same treatment in KEYNOTE-189 (39% vs 48%, respectively)[15]. The ORR for pembrolizumab plus chemotherapy in PERLA was also lower in the PD-L1 TPS 1–49% and ≥50% subgroups compared with the same subgroups in KEYNOTE-189 (34% vs 49% and 48% vs 62%, respectively), while the ORR in the PD-L1-negative subgroup was consistent between these trials[15]. The ORR for dostarlimab plus chemotherapy in PERLA was comparable to pembrolizumab plus chemotherapy in the KEYNOTE-189 trial across subgroups[15]. There was a smaller proportion of patients in the dostarlimab plus chemotherapy arm with unknown or missing responses ('not done') than in the pembrolizumab plus chemotherapy arm (4% and 11%, respectively) in the current study; this was primarily due to a greater number of patients in the pembrolizumab plus chemotherapy arm with no post-baseline disease assessments due to death from a fatal AE (7%) compared with dostarlimab plus chemotherapy (2%) (Supplementary Table 4). This could have negatively impacted the ORRs for the pembrolizumab plus chemotherapy treatment group in PERLA relative to the ORRs reported in KEYNOTE-189,

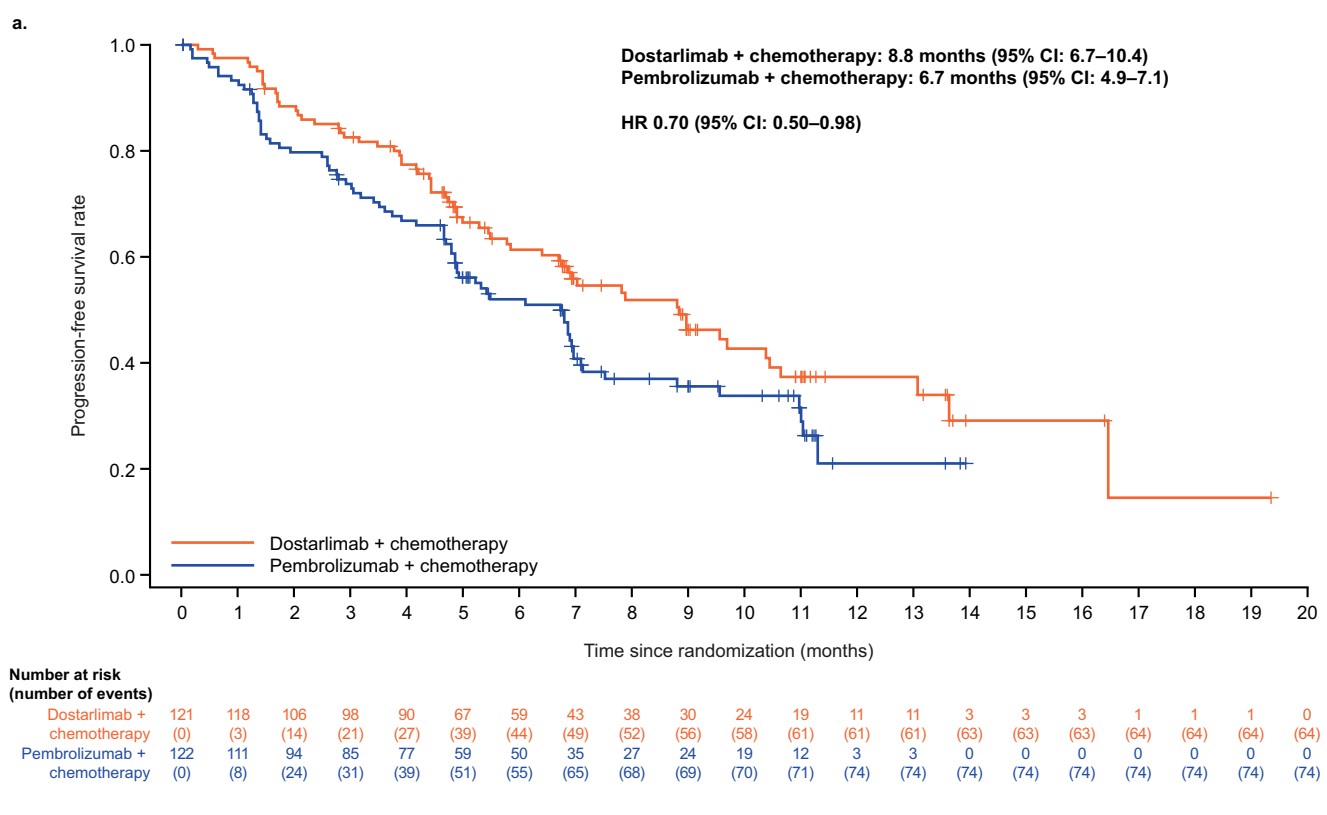

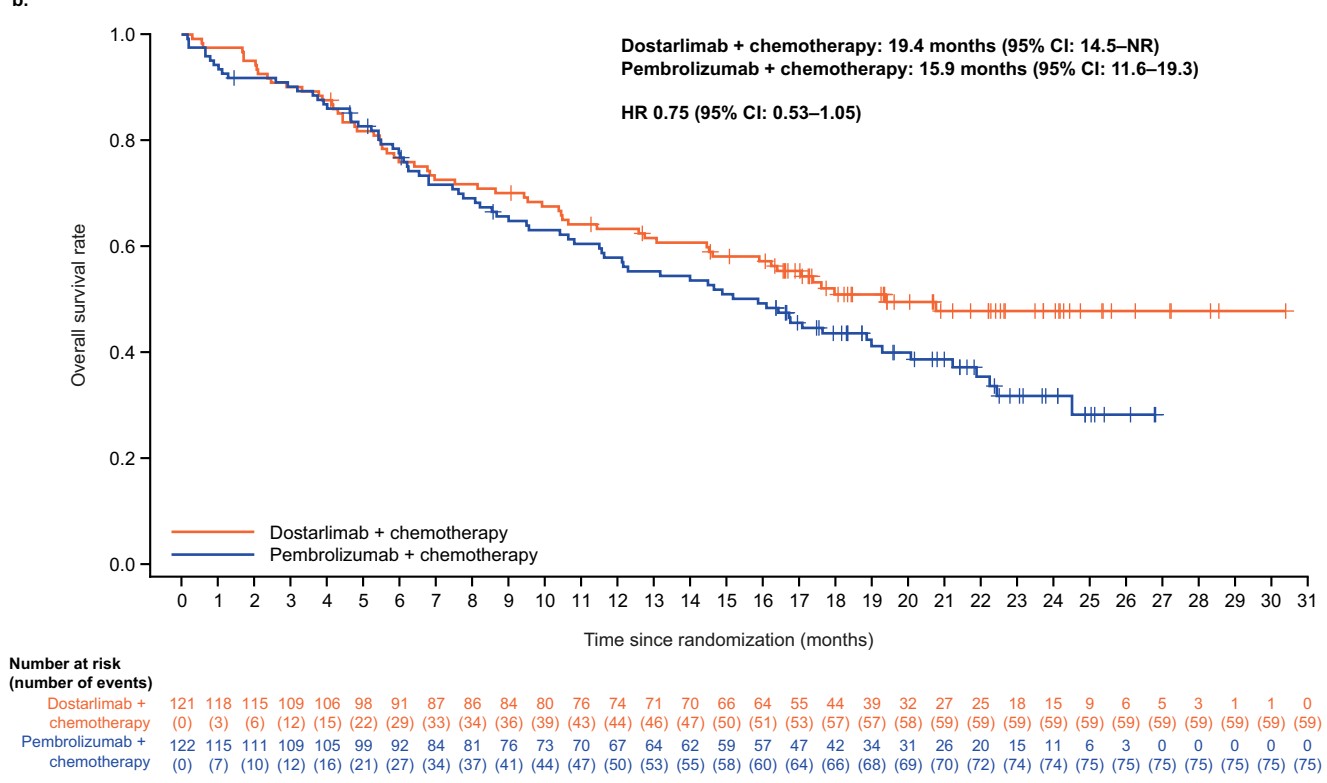

**Fig. 3 | Kaplan–Meier PFS and OS curves of dostarlimab plus chemotherapy and pembrolizumab plus chemotherapy. a** PFS (data cut-off: August 4, 2022) and **b** OS (data cut-off: July 7, 2023) Note: + symbols represent individual censoring events. CI confidence interval, HR hazard ratio, NR not reached, OS overall survival, PFS progression-free survival.

**Table 4 | OS for patients receiving dostarlimab + chemotherapy and pembrolizumab + chemotherapy (ITT population as of July 7, 2023)**

| Variable | Dostarlimab + chemotherapy (N = 121) | Pembrolizumab + chemotherapy (N = 122) |
|---|---|---|
| Median OS follow-up time, months (IQR) | 20.7 (17.3–24.0) | 21.6 (18.3–24.1) |
| OS events observed, n | 59 | 75 |
| Median OS (95% CI), months | 19.4 (14.5–NR) | 15.9 (11.6–19.3) |
| Estimated probability of OS, % (95% CI) | | |
| 6 months | 76 (67–83) | 78 (69–84) |
| 12 months | 63 (54–71) | 58 (48–66) |
| Hazard ratio (95% CI) | 0.75 (0.53–1.05) | |

*CI* confidence interval, *IQR* interquartile range, *ITT* intention-to-treat, *NR* not reached, *OS* overall survival.

**Table 5 | Overall summary of AEs (safety population as of July 7, 2023)**

| AE (n, %) | Dostarlimab + chemotherapy (N = 121) | Pembrolizumab + chemotherapy (N = 122) |
|---|---|---|
| AEs | 119 (98) | 119 (98) |
| TRAEs | 103 (85) | 99 (81) |
| Dostarlimab or pembrolizumab-related AEs | 86 (71) | 70 (57) |
| Grade ≥3 AEs | 78 (64) | 78 (64) |
| Grade ≥3 TRAEs | 78 (64) | 77 (63) |
| AEs leading to treatment discontinuation | 35 (29) | 46 (38) |
| Dostarlimab or pembrolizumab-related AEs leading to treatment discontinuation | 20 (17) | 29 (24) |
| SAEs | 50 (41) | 58 (48) |
| Dostarlimab or pembrolizumab-related SAEs | 14 (12) | 17 (14) |
| Fatal AEs[a] | 15 (12) | 12 (10) |
| Fatal TRAE[b] | 3 (2) | 5 (4) |
| Immune-mediated lung disease | 1 (<1) | 0 |
| Myelosuppression | 0 | 1 (<1) |
| Pneumonia | 0 | 1 (<1) |
| Pneumonitis | 1 (<1) | 0 |
| Septic shock | 0 | 1 (<1) |
| Thrombocytopenia | 0 | 1 (<1) |
| Fatal dostarlimab or pembrolizumab-related AEs | 3 (2)[c] | 2 (2)[d] |
| irAEs[e] | 38 (31) | 47 (39) |
| irSAEs | 12 (10) | 11 (9) |

*AE* adverse event, *ir* immune-related, *SAE* serious adverse event, *TRAE* treatment-related adverse event.
[a]Patients who had a fatal AE recorded and death was not recorded as due unequivocally to disease under study.
[b]AEs described as treatment-related could be related to any study treatment agent.
[c]Pneumonitis; immune-related pneumonitis, urosepsis.
[d]Myelosuppression, respiratory failure.
[e]No new immune-related deaths were observed.

where fewer patients (3%) had no assessment[15]. However, subgroup analyses by PD-L1 status showed that the largest discrepancy in the proportion of patients with unknown or missing ('not done') responses was in the TPS < 1% subgroup (2% for dostarlimab plus chemotherapy vs 12% for pembrolizumab plus chemotherapy); this was the only PD-L1 subgroup in which ORR did not numerically favor dostarlimab.

A key difference in patient characteristics between KEYNOTE-189 and PERLA is the slightly higher percentage of patients with brain or liver metastases who received pembrolizumab plus chemotherapy in KEYNOTE-189 (18% and 16%, respectively)[15] than in PERLA (12% and 11%, respectively). In KEYNOTE-189, the presence of liver or brain metastases was associated with numerically shorter median PFS and OS[15]. Future analyses of additional data collected in PERLA, such as pharmacokinetic and pharmacodynamic endpoints, may help to further explain any discrepancies between dostarlimab and pembrolizumab in this study. The ORR of 46% for patients receiving dostarlimab plus chemotherapy was also comparable to that for patients with advanced (Stage IV) non-squamous NSCLC receiving the anti-PD-L1 antibody atezolizumab plus chemotherapy (carboplatin plus nanoparticle albumin-bound paclitaxel) as 1 L treatment in the IMpower130 study (49%)[16]. In PERLA, both dostarlimab plus chemotherapy and pembrolizumab plus chemotherapy demonstrated antitumor activity in PD-L1-negative patients as well as in those who were PD-L1-positive.

PFS and OS were also similar between patients treated with dostarlimab plus chemotherapy and those treated with pembrolizumab plus chemotherapy. PFS (data cut-off: August 4, 2022) was broadly consistent across PD-L1-positive subgroups, with some small differences observed. The median PFS of 8.8 months for dostarlimab plus chemotherapy was numerically higher versus pembrolizumab plus chemotherapy (6.7 months) in the current study and compared favorably to that observed in patients treated with pembrolizumab plus chemotherapy in KEYNOTE-189 (9.0 months)[15]. Interestingly, the PFS observed in patients treated with pembrolizumab plus chemotherapy in PERLA is similar to that observed in the real-world setting (6.7 and 5.9 months, respectively)[17], suggesting that the population analyzed in PERLA is representative of real-life clinical practice.

OS data from the August 4, 2022, data-cut were immature (37% maturity, with 90 events in a total of 243 patients; Supplementary Table 8). A pre-planned updated analysis of the July 7, 2023 data cut provided longer survival follow-up and more mature OS data, and demonstrated a numerical trend in median OS favoring dostarlimab plus chemotherapy versus pembrolizumab plus chemotherapy (19.4 months vs 15.9 months, respectively); separation of the OS curves for dostarlimab plus chemotherapy and pembrolizumab plus chemotherapy appears to start from 6 months as per the Kaplan-Meier curves. The pembrolizumab plus chemotherapy arm in PERLA had a lower median OS than the same arm in KEYNOTE-189 (15.9 months and 22.0 months,

respectively)[18]. Importantly, recent evidence consistently demonstrates that the effectiveness of pembrolizumab, and other anti-PD(L)−1 combinations with chemotherapy, is lower in the real-world than what is reported in the KEYNOTE-189 trial or other randomized trials[17]. Randomized trials demonstrate a significant variability in median OS data for chemotherapy and immunotherapy combinations, suggesting patient selection, variability in standards of care over time, and geographical locations play a role in this variability. KEYNOTE-189 is characterized to date as the best OS reported (median follow-up 23.1 months and 5-year readout median OS 22.0 months)[15,18], significantly higher than subsequent trials (median follow-up: 16.3–34.9 months; median OS: 13.3–21.9)[19–21] and real-world data (median OS: 5.1–16.2)[17]—emphasizing the strict limitation in intertrial comparisons.

The evidence of a delayed treatment effect for dostarlimab plus chemotherapy was more prominent in the PD-L1 TPS ≥1% subgroup (Supplementary Fig. 4); OS Kaplan–Meier curves overlapped in the PD-L1 TPS < 1% subgroup. Median OS for the pembrolizumab plus chemotherapy subgroup in PERLA was consistent with the same subgroup in KEYNOTE-189 (16.1 months and 17.2 months, respectively)[18]. Exploratory analysis of DOR (Supplementary Fig. 1) showed a similar DOR for dostarlimab plus chemotherapy and pembrolizumab plus chemotherapy groups, based on a small subset of the study population (103 patients in total).

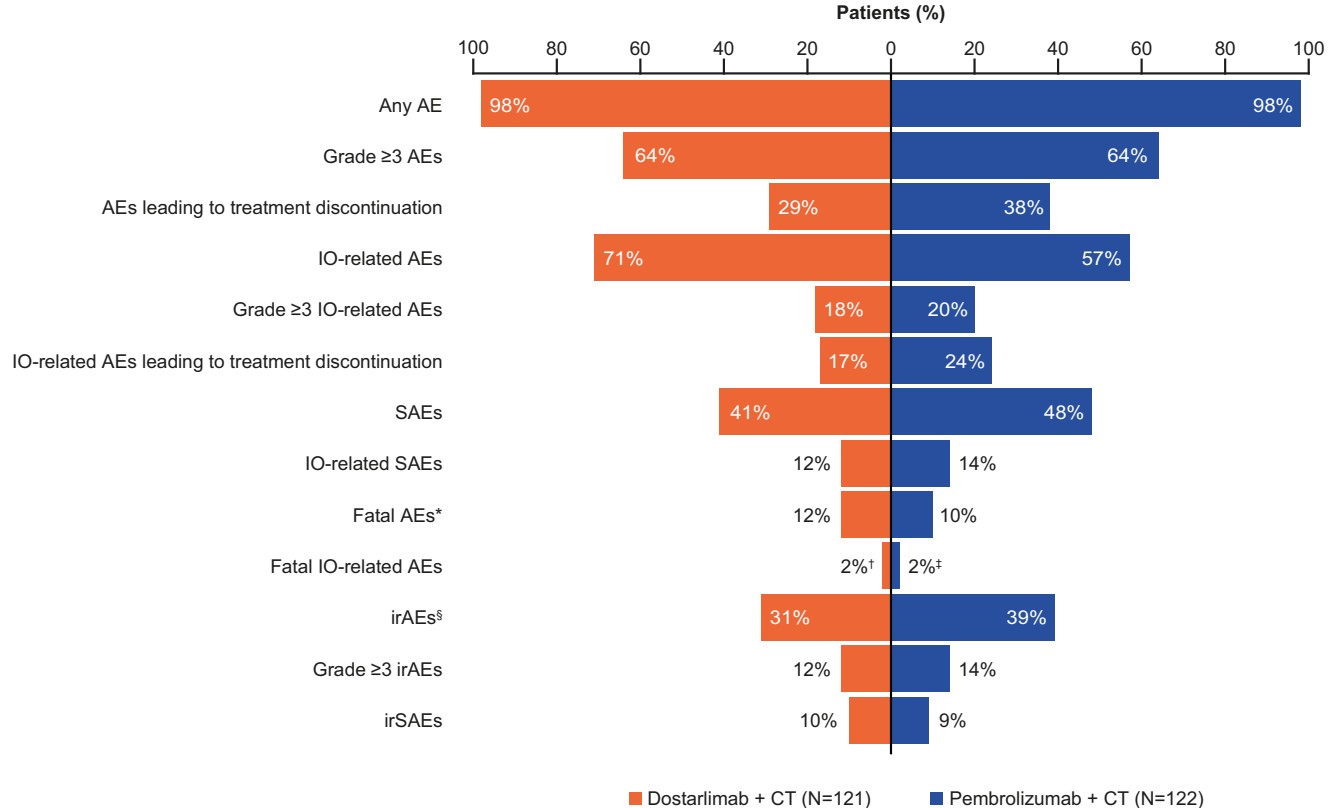

**Fig. 4 | Tornado plot showing types of AEs in patients receiving dostarlimab + chemotherapy and pembrolizumab + chemotherapy (safety population as of July 7, 2023).** *Subjects who had a fatal AE recorded and died were not recorded as due unequivocally to disease under study. †Pneumonitis; immune-related pneumonitis, urosepsis. ‡Myelosuppression, respiratory failure. §No new immune-related deaths were observed. AE adverse event, CT chemotherapy, IO immuno-oncology agent, ir immune-related, SAE serious adverse event.

Safety results were consistent with previously published data on PD-1 inhibitors, with no new safety signals identified. In both treatment groups, 98% of patients experienced any TEAE; in the pembrolizumab plus chemotherapy arm of KEYNOTE-189 this was 99.8%[15]. It was noted that several safety parameters numerically favored dostarlimab across both data cuts, including the proportion of patients experiencing SAEs, irAEs, and AEs leading to treatment discontinuation, while dostarlimab-related AEs were higher than pembrolizumab-related AEs.

Overall, the antitumor activity of dostarlimab plus chemotherapy in PERLA is in line with previously published data on other 1 L PD-1 inhibitor-chemotherapy combinations in similar non-squamous NSCLC patient populations. Although studies are not directly comparable due to differences in design and populations, the current findings support the hypotheses that dostarlimab plus chemotherapy is effective and tolerable in patients with metastatic non-squamous NSCLC, and that dostarlimab is a suitable PD-1 inhibitor backbone for future combinations under investigation.

There is interest among the scientific and clinical community in the comparability of PD-1 inhibitors and whether they show clinically relevant intra-class differences. Cross-trial comparisons suggest that there may be differences in safety[22], and comparison of pharmaco-kinetic and pharmacodynamic parameters such as binding site, affinity, and half-life have also shown a degree of variability[23]. However, PERLA represents the first large global study to compare two PD-1 inhibitors head-to-head in the same indication. While this study was not powered to assess superiority, the two-sided 80% CI for the difference in ORR between dostarlimab and pembrolizumab does not cross 0, suggesting that there may be clinically relevant differences between the two PD-1 inhibitors. As discussed above, future analyses of pharmacodynamic and pharmacokinetic data will provide further insight into these differences and their potential relevance for clinical practice.

This study has several strengths and limitations. Strengths include the use of BICR to assess response rates for the primary endpoint of ORR and the global nature of the study, which allowed a diverse patient population to be recruited, including patients from East Asia, South America, the US, and Europe. Global recruitment was facilitated using dynamic eligibility criterion regarding genomic aberrations. In terms of limitations, this is a Phase II study with a small sample size that was not designed to be powered to statistically confirm superiority. While the study design was based on a non-inferiority trial, the large non-inferiority margin and type I error rate (dictated by consideration of the feasible sample size) precluded a hypothesis of true non-inferiority. As such, the pre-specified hypothesis was that the two treatments were 'similar'. This limits inferences of differences between the two PD-1 inhibitors.

In conclusion, dostarlimab plus chemotherapy demonstrates strong clinical efficacy, similar to pembrolizumab plus chemotherapy, in 1 L metastatic non-squamous NSCLC. Safety profiles were also similar and were consistent with published data in PD-(L)1 inhibitors. These results support the further investigation of dostarlimab as a backbone for future use in combination with standard of care and future novel cancer therapies.

## Methods
### Ethics and approval
All patients provided written informed consent before participation in the study, which was conducted in accordance with the Declaration of Helsinki[24] and International Ethical Guidelines, International Council for Harmonisation Good Clinical Practice guidelines, and all local laws.

The study was overseen by an internal safety monitoring committee, comprised of GSK employees who were not part of the dostarlimab development program. The protocol (available at: https://www.gsk-studyregister.com/en/trial-details/213403) was approved by relevant ethics committees and institutional review boards at each study site (Supplementary Table 1). After trial commencement, eligibility criteria were amended to allow patients with a recent history of a wider range of minimally invasive cancers; this change was made in response to a subject with a previous malignancy that the study team agreed should have been eligible. Additional amendments to the original protocol are listed with rationale in Supplementary Table 2.

## Study design and participants

PERLA (NCT04581824) was a global, multicenter, Phase II, randomized, double-blind, parallel, two-arm trial (Fig. 1), which aimed to enroll ~240 patients (120 per treatment group). The trial was conducted at 54 study sites across 12 countries (Republic of Korea, Taiwan, France, Germany, Italy, Poland, Romania, Spain, Argentina, Brazil, Chile, and the US). Patients were not compensated for participation in the study, except for reimbursement for travel expenses. The first patient was enrolled on November 19, 2020, and the last patient on February 18, 2022. Eligible patients were ≥18 years of age, with metastatic non-squamous NSCLC, measurable disease (per RECIST v1.1 criteria, investigator-assessed) and did not have genomic aberrations for which an approved targeted therapy was regionally available. Genomic aberrations were locally assessed, with the exact list depending on regional approvals. Eligible patients also had documented PD-L1 status assessed by local or central testing using a 22C3 pharmDx assay (Agilent/Dako), ECOG performance status score of 0–1, and a life expectancy ≥3 months. Patients were excluded if they had received prior systemic therapy for metastatic NSCLC or prior therapy with a PD-(L)1/2 inhibitor or another immunotherapy agent. Patient sex was self-reported. Complete inclusion/exclusion criteria are presented in Supplementary Table 3.

## Randomization and blinding

Eligible patients were randomized 1:1 via blocking using an interactive web response system (RAMOS NG), to receive chemotherapy in combination with either dostarlimab or pembrolizumab. Patients, relevant study staff, and investigators were blinded to study treatment; details of blinding were documented in site-specific blinding plans. Randomization was stratified by PD-L1 status (TPS < 1% vs 1–49% vs ≥50%) and smoking status (never vs former/current).

## Interventions and assessments

Patients received either 500 mg dostarlimab or 200 mg pembrolizumab, administered as a 30-minute intravenous (IV) infusion, every 3 weeks (Q3W) for up to 35 cycles (~24 months) or until disease progression, withdrawal of consent, unacceptable toxicity, or death. Patients also received pemetrexed (500 mg/m$^2$ IV Q3W) for up to 35 cycles, together with either cisplatin (75 mg/m$^2$ IV Q3W) or carboplatin (area under the curve 5 mg/mL/min IV Q3W) for the first four cycles, based on investigator discretion. The pembrolizumab plus chemotherapy arm was similar to the intervention described in the KEYNOTE-189 study[25]. Dose modifications were not permitted for dostarlimab or pembrolizumab during the study, although treatment could be withheld for up to 12 weeks for AE management. Chemotherapy dose modifications were permitted, based on investigator judgement and local product label recommendations.

During treatment, clinical assessments of tumor responses (per RECIST v1.1, by BICR and by the Investigator; treatment decisions were based on investigator assessment) were supported by serial imaging (computed tomography or magnetic resonance imaging) carried out on Weeks 6 and 12, then every 9 weeks up to Week 48, and every 12 weeks thereafter until discontinuation of study treatment. Other regions (e.g., brain) were imaged at baseline, as clinically indicated, by computed tomography or magnetic resonance imaging.

Safety (monitoring of AEs) was assessed throughout the treatment period and at visits occurring 30 and 90 days after the last dose of study treatment. Post-treatment follow-up assessments occurred 180 days after the last dose of study treatment and every 90 days thereafter until death, withdrawal of consent, or end of study.

## Outcomes

The primary endpoint was confirmed ORR, as measured by BICR per RECIST v1.1 criteria, defined as the proportion of patients with a best overall response of either CR or PR. ORR was assessed after all patients had completed the third on-study tumor assessment (after ~6 months) or had discontinued from the study, whichever occurred first. Patients with unknown or missing responses were counted as 'not done' but were included in the denominator when calculating percentage of responses.

Secondary efficacy endpoints included PFS (defined as the time from date of randomization to date of disease progression [as per RECIST v1.1 by investigator assessment] or any-cause death, whichever occurred first) and OS (defined as the time from date of randomization to the date of any-cause death). Pre-specified exploratory analyses of ORR and PFS by PD-L1 subgroups TPS < 1%, TPS 1–49%, TPS ≥ 50%, and TPS ≥ 1% were also performed. A prespecified subgroup analysis of ORR by sex was conducted and was previously reported[26]. Prespecified exploratory analyses of DOR (defined as the time from first documented CR or PR until documented disease progression [as per RECIST v1.1 by BICR], or death, whichever occurs first) was also performed.

Safety assessments included vital signs, clinical laboratory parameters, and incidence of TEAEs, SAEs, TEAEs leading to death, and AEs leading to discontinuation. AEs were coded using standard Medical Dictionary for Regulatory Activities (MedDRA) terminology and graded by the investigator according to the National Cancer Institute Common Terminology Criteria for Adverse Events (NCI-CTCAE) v5.0.

Two data-cuts have been used for the data analyses in this manuscript. ORR, PFS and safety analyses have been reported per August 4, 2022, data cut-off. Updated analyses of ORR, DOR, OS and safety have been reported per July 7, 2023, data cut-off.

## Data collection and statistical analysis

The planned sample size for the study was 240 (~120 patients in each arm), providing 85% power to show that ORR for dostarlimab plus chemotherapy is not more than 15% worse than pembrolizumab plus chemotherapy at the 10% one-sided level of significance, assuming the true ORR was 45% for both treatment groups. 'Similarity' was implied as the ORR for dostarlimab plus chemotherapy not being >15% lower than the ORR for pembrolizumab plus chemotherapy.

Efficacy analyses (stratified by PD-L1 and smoking status) were performed using the intent-to-treat (ITT) population, defined as all patients randomized to study treatment. Safety analyses were performed using the safety population, defined as all patients randomized to study treatment who received at least one dose of study medication.

Data were collected using Veeva EDC (CDMS) versions 20R3, 21R1, 21R3, 22R1, 22R2 and 22R3 and analyzed using SAS software version 9.4. Baseline demographics and characteristics for the ITT population are presented using descriptive statistics and safety outcomes (AEs) are presented by frequency and proportion. For the primary endpoint (ORR by BICR), the Mantel and Haenszel method with Sato's variance estimator was used for treatment comparison (stratified by PD-L1 status and smoking status); the difference in ORR between groups and

its 80% CI are reported. The Clopper–Pearson method was used to calculate point estimates for ORR, with 95% CIs. ORR in patients with PD-L1 TPS ≥ 1% was included as a prespecified subgroup analysis, in addition to the PD-L1 stratification subgroups TPS < 1%, 1–49%, and ≥50%.

PFS and OS were analyzed using Kaplan–Meier methodology. PFS and OS HRs and 95% CIs were estimated using a Cox proportional hazards model with PD-L1 status and smoking status as stratification factors. Maturity for time-to-event analyses was calculated as a percentage of patients in the ITT population (PFS and OS) or as a percentage of the subset of patients with CR or PR (duration of response) that had experienced an event at data cut-off. Duration of follow-up was calculated using the Kaplan–Meier estimate of potential follow-up[27]. Time from first and last patient randomized to the time of data cut-off was also calculated. There were no imputation methods used for missing data.

### Reporting summary

Further information on research design is available in the Nature Portfolio Reporting Summary linked to this article.

## Data availability

Clinical data are available under restricted access for confidentiality reasons; researchers can request access to our studies by providing a scientific research proposal with a commitment to publish their findings. Researchers whose requests are approved by an independent panel and accepted by GSK are provided access to data in a secure environment upon signing a Data Sharing Agreement (DSA). Review criteria for research proposals are: Scientific rationale and relevance of the proposed research to medical science or patient care, ability of the proposed research plan (design, methods and analysis) to meet the scientific objectives, qualifications and experience of the research team to conduct the proposed research review, whether the proposal has potential to produce information that may increase the risk of identification of individual research participants, any real or potential conflicts of interest that may impact the planning, conduct or interpretation of the research and proposals to manage these conflicts of interest, and the publication plan for the research. In addition, patients give permission through an informed consent form to use their data for original studies, so further research must study the medicine or disease that was researched in the original studies. Data will not be provided to requesters where there is a potential conflict of interest, data are to be used for a commercial purpose or there is an actual or potential competitive risk, and researchers are required to sign a DSA, which includes requirements to publish results of the analysis in a scientific journal or pre-print option and open-source release of any software or models. Submitted proposals will be acknowledged within a week and anonymized data will be shared within 30 days of signing the DSA. Access to data and documents is provided for 12 months with the possibility of extension up to an additional 6 months. Please see https://www.gsk-studyregister.com/About_GSK_Patient_Level_Data_Sharing_Final_13July2023.pdf for full details. Access criteria are correct as of August 2023 – the latest information will be available at https://www.gsk-studyregister.com/en/. GSK is committed to share anonymized subject level data from interventional trials as per GSK policies (https://www.gsk.com/en-gb/innovation/trials/data-transparency/) and as applicable. The raw individual participant data are protected and cannot be made publicly available (as source data) due to data privacy laws. The anonymized individual participant data can be requested for further research at https://www.gsk-studyregister.com/en/. The study documents (including the study protocol and statistical analysis plan) are available at https://www.gsk-studyregister.com/en/trial-details/?id=213403. The remaining data are available within the Article and its Supplementary Information.

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

## Acknowledgements

Editorial support was provided by Eva Kane, PhD, and Mary-Clare Cathcart, PhD, of Fishawack Indicia Ltd., part of Avalere Health, and was funded by GSK. This study was funded by GSK (213403; NCT04581824). The trial was designed by GSK, in collaboration with the authors. The sponsor provided support for the statistical analyses of the data and funded a medical writer for the report. Authors performed the data collection, analysis, and interpretation of the data. The authors had the final decision to submit the manuscript for publication. The manuscript was written by the authors with medical writing assistance funded by GSK. GSK had a role in the design and conduct of the study; collection, management, analysis, and interpretation of the data; preparation, review, or approval of the manuscript; and decision to submit the manuscript for publication. GSK collaborated with the investigators in designing the trial, provided the study drug, coordinated the management of the study sites, funded the statistical analysis, and provided medical writing support. Authors employed by GSK, in coordination with all authors, were involved in preparation, review, approval, and decision to submit the manuscript.

## Author contributions

S.M.L., A.L.O.G., G.d.J.P., C.S.F., G.L.R., J.S.A., M.S., and F.d.M. contributed to data acquisition; S.P., M.R., E.B., N.S., and Z.S. contributed to data analysis/interpretation. S.O'.D., E.Z., and N.H. contributed to study conception/design and data analysis/interpretation. All authors contributed to drafting and critical revision of the manuscript.

## Competing interests

S.M.L. has received research grants from Yuhan and Janssen; received consulting fees from AstraZeneca, Boehringer Ingelheim, Lilly, Takeda, Guardant and J Ints Bio; and received research grants from AstraZeneca, Beigene, Boehringer Ingelheim, GSK, Roche, Hengrui, BridgeBio Therapeutics, Oscotec, and Daichii-Sankyo. S.P. has an advisory role with AbbVie, AiCME, Amgen, Arcus, AstraZeneca, Bayer, Beigene, Biocartis, BioInvent, Blueprint Medicines, Boehringer Ingelheim, Bristol-Myers Squibb, Clovis, Daiichi Sankyo, Debiopharm, ecancer, Eli Lilly, Elsevier, F-Star, Fishawack, Foundation Medicine, Genzyme, Gilead, GSK, Illumina, Imedex, IQVIA, Incyte, iTeos, Janssen, Medscape, Medtoday, Merck Sharp and Dohme, Merck Serono, Merrimack, Novartis, Novocure, OncologyEducation, Pharma Mar, Phosplatin Therapeutics, PER, Peerview, Pfizer, PRIME, Regeneron, RMEI, Roche/Genentech, RTP, Sanofi, Seattle Genetics, Takeda, and Vaccibody; has been an invited speaker for AiCME, AstraZeneca, Boehringer Ingelheim, Bristol-Myers Squibb, ecancer, Eli Lilly, Foundation Medicine, Illumina, Imedex, Medscape, Merck Sharp and Dohme, Mirati, Novartis, PER, Peerview, Pfizer, Prime, Roche/Genentech, RTP, Sanofi, and Takeda; and received research grants from Amgen, AstraZeneca, Beigene, Bristol-Myers Squibb, GSK, Merck Sharp and Dohme, and Roche/Genentech. A.L.O.G. is an employee of the Servicio Andaluz de Salud and has had an advisory role for Roche, Bristol Myers Squibb, and Merck Sharp Dohme. GdJP has no conflicts of interest to disclose. C.S.F. has been an invited speaker for Fundacion Respirar. G.L.R. has received consulting fees from Roche, Novartis, BMS, MSD, AstraZeneca, Takeda, Amgen, Sanofi, Italfarmaco and Pfizer; received honoraria from Roche, Novartis, BMS, MSD, AstraZeneca, Takeda, Amgen, and Sanofi; received travel grants from Roche, BMS, and MSD; had an advisory role for Roche, Novartis, BMS, MSD, AstraZeneca, and Sanofi; and been an investigator for clinical trials sponsored by Roche, Novartis, BMS, MSD, AstraZeneca, GSK, Amgen, and Sanofi. MS has had contracts for clinical trial activities (institutional and personal as site Principal Investigator) with GSK, Merck Serono, BMS, MSD, Roche, Sanofi, Regeneron, Astellas, Amgen, Bayer, BeiGene, Clovis, Tesaro, Gilead, Bioven, Novartis, Pfizer, Eli Lilly, Pharma Mar, AbbVie, Astra Zeneca, Mylan, and Daiichi Sankyo. JSA has been an invited speaker for Kyowa Kirin, Amgen Korea, Yuhan, AstraZeneca Korea, Menarini Korea, Bayer Korea, Takeda Phar, Novartis Korea, Hanmi, BC World, Pfizer, Roche Korea, and Boehringer Ingelheim; and had an advisory role for Yuhan, Bayer Korea, Yooyoung, Pharmbio Korea, Vifor Pharma, and Bixink. MR has had an advisory role for Amgen, AstraZeneca, Beigene, BMS, Boehringer Ingelheim, Daiichi-Sankyo, GSK, Lilly, Merck, MSD, Novartis, Pfizer, Regeneron, Roche, Samsun Bioepsis, and Sanofi. Z.S. is an employee of GSK. N.H. is a former employee of GSK (at time of study) and is a current employee of Lantheus Medical Imaging. E.Z. is an employee of GSK. E.B. is an employee of GSK and holds stocks/shares in GSK. N.S. is an employee of GSK and their spouse works for Debiopharm. S.O'.D. is an employee of GSK and their spouse works as a federal employee. F.d.M. has had an advisory role for AstraZeneca, Roche, Novartis, Merck, BMS, and MSD.

## Additional information

Sun Min Lim ®[1,16] ✉, Solange Peters ®[2,16], Ana Laura Ortega Granados ®[3], Gustavo dix Junqueira Pinto[4], Christian Sebastián Fuentes[5], Giuseppe Lo Russo[6], Michael Schenker[7], Jin Seok Ahn[8], Martin Reck[9], Zsolt Szijgyarto[10], Neda Huseinovic[11,15], Eleftherios Zografos[10], Elena Buss[12], Neda Stjepanovic[12], Sean O'Donnell[13] & Filippo de Marinis[14]

[1]Division of Medical Oncology, Department of Internal Medicine, Yonsei Cancer Center, Severance Hospital, Yonsei University College of Medicine, 50-1 Yonsei-ro, Seodaemun-gu, Seoul 03722, South Korea. [2]Oncology Department, Centre Hospitalier Universitaire Vaudois, Lausanne University, Rue du Bugnon 21, 1011 Lausanne, Vaud, Switzerland. [3]Medical Oncology Department, Hospital Universitario de Jaén, Avda. Del Ejército Español 10, 23007 Jaén, Spain. [4]Department of Medical Oncology, Barretos Cancer Hospital, Rua Antenor Duarte Villela, 1331 Bairro Dr. Paulo Prata, Barretos, São Paulo 14784-400, Brazil. [5]Centro de Investigaciones Clínicas, FUNDACIÓN RESPIRAR, Av. Cabildo 1548 1er, Buenos Aires, Argentina. [6]Medical Oncology Department 1, Thoracic Unit, Fondazione IRCCS Istituto Nazionale dei Tumori, Via Giacomo Venezian, 1 20133 Milan, Italy. [7]Sf Nectarie Oncology Center, 23 Strada Caracal, Craiova, Județul Dolj, Romania, and the University of Medicine and Pharmacy, Craiova, Romania. [8]Division of Hematology-Oncology, Department of Medicine, Samsung Medical Center, Sungkyunkwan University, 81 Irwon-ro, Gangnam-gu, Seoul 06351, South Korea. [9]Lungen Clinic, Airway Research Center North, Center for Lung Research, Wöhrendamm 80, 22927 Grosshansdorf, Germany. [10]GSK, Gunnels Wood Road, Stevenage SG1 2NY, UK. [11]GSK, 1000 Winter Street, Waltham, MA 02451, USA. [12]GSK, Neuhofstrasse 4, 6340 Baar (Zug), Switzerland. [13]GSK, 1250 S Collegeville Rd, Collegeville, PA 19426, USA. [14]Division of Thoracic Oncology, Istituto Europeo di Oncologia (IRCCS), Via Ripamonti 435, Milan, Italy. [15]Present address: Lantheus Medical Imaging, 201 Burlington Road, Bedford, MA 01730, USA. [16]These authors contributed equally: Sun Min Lim, Solange Peters. ✉e-mail: LIMLOVE2008@yuhs.ac

