## [Peer Review File · Nature Communications]

Dostarlimab or Pembrolizumab Plus Chemotherapy in Previously Untreated Metastatic Non-Squamous Non-Small Cell Lung Cancer: the Randomized PERLA Phase II TrialEditorial Note: Parts of this Peer Review File have been redacted as indicated to maintain patient confidentiality.

REVIEWER COMMENTS

Reviewer #1 (Remarks to the Author): with expertise in clinical trial study design, biostatistics

Thank you for inviting me to review the manuscript "Randomized Phase II Trial: Dostarlimab or Pembrolizumab Plus Chemotherapy in Previously Untreated Metastatic Non-Squamous Non-Small Cell Lung Cancer" by Lim et al. I reviewed the manuscript with a focus on statistical analysis and study design.

While in general the manuscript is well written and the results are presented clearly, I have some concerns regarding some of the design aspects and the conclusions.

In the study protocol and the manuscript, it is mentioned, that the aim was to investigate "similarity" between the study groups. Nevertheless, according to the sample size calculation and the methods used, the study was designed to test for superiority of one treatment option over the other. If the primary goal was to investigate non-inferiority of one treatment as compared to the other or equivalence between the treatment options, this should have been addressed in the planning phase and in the study protocol. Generally, "similarity" does not appear to be a well-defined term. Additionally, in the sample size calculation a one-sided test is mentioned, but I was not able to find the pre-specified direction of the hypotheses to be tested.

In the study protocol and the manuscript, a sample size calculation is presented. For the sample size calculation, a "10% one-sided type I error rate" was used and sample size was determined to allow detection of a difference in ORR of 15 percentage points between the groups with appropriate power.

While the chosen significance level is very liberal, I think it is acceptable to use it in a phase II randomized trial.

Despite the clear and proper sample size calculation it is mentioned in the discussion that the "study was not formally designed to demonstrate, or statistically powered to test for, superiority". This should be clarified or discussed in more detail.

In the primary analysis a difference in ORR of 9 percentage points was observed. This is smaller than the 15 percentage points considered in the sample size calculation, but lead to a statistically significant difference between the groups (no p values are shown, but the 80% confidence interval for the difference in ORR between the groups does not cover the value of zero). For PFS a hazard ratio of 0.70 and difference in median event times of two months were observed. While it is a matter of debate, whether these observed differences can be interpreted as clinically relevant differences between the study groups, I believe that this should be discussed in more detail. In my opinion, just stating that both groups showed similar results oversimplifies the discussion of the design aspects and the results observed.

Minor issues:

Quantification of follow-up times should be reconsidered (see Schemper & Smith, Controlled Clinical Trials, 1996).

Number of decimals should be presented more consistently (for example same number of decimals for point estimates and confidence intervals or for means and standard deviations).

For subgroup analyses, test on interactions should be added to investigate heterogeneity of treatment effects.

Table 2 (ORR by PD-L1 TPS subgroup) is not well formatted and therefore difficult to read.

In Table 3, absolute and relative frequencies of PFS events are shown. While absolute numbers are fine, relative frequencies should be deleted, as these might be misleading due to different lengths of follow-up.

Table 4 is not well labelled. Header of the first column is "Median PFS", but also subgroup sizes and hazard ratios are shown.

Bernhard Haller

Biostatistician

Institute for AI and Informatics in Medicine

Technical University of Munich

Reviewer #2 (Remarks to the Author): with expertise in lung cancer, (immuno)therapy

Congratulations on completing the first comparative chemo-IO study using what is broadly considered a standard of care.

The inclusion of patients based on local testing is to be applauded. Was any central reproducibility testing done?

How do the authors anticipate using this data as it was not designed for superiority study or formal non-inferiority?

Please reformat the tables for readability, the CI are not tracking on line with the data.

Can the grade 5 event of myelosuppression be more appropriately coded? Is this aplastic anemia? Myelosuppression is not a CTCAE term.

Reviewer #3 (Remarks to the Author): with expertise in lung cancer, (immuno)therapy

In the manuscript from Lim et al., the authors report the primary outcomes from a randomized, double-blind, international phase II trial comparing two different anti-PD-1 antibodies in combination with chemotherapy for newly diagnosed patients with metastatic

non-squamous NSCLC. Although a phase II trial, the investigators should be commended for completing this international double-blind trial with 243 patients, which represents the first to directly compare two anti-PD-1 antibodies. The manuscript is well-written, easy to read, with clear presentation of the data and careful interpretations of the findings. The authors have been upfront about the trial design, statistical considerations and inherent limitations. As such, they are clear to point out that the trial was not designed to test the superiority of dostarlimab vs pembrolizumab, and that the results instead demonstrate similar performance in the efficacy and toxicity profiles of the two regimens. They have further done a good job in the Discussion to place the results in the context of other trials (e.g., including the fact that the control arm with pembrolizumab underperformed vs. the KN-189 results) and current international standards of care. Overall, the work is very timely and of considerable interest to both the lung cancer and broader oncology communities, as multiple similar PD-(L)1 inhibitors are being developed for clinical use, especially in combination therapy regimens.

I have a few specific comments/suggestions for consideration:

1. The authors do not outline the rates of usage of cisplatin vs carboplatin amongst the patients. Although this is unlikely to significantly change the results in any substantive way, given the international nature of the trial, it could give hints as to regional differences in practice patterns or outcomes.
2. The authors indicate in the study design that eligible patients were defined as not having “genomic alterations for which an approved targeted therapy was regionally available.” This is a practical, real-world consideration that I think is appropriate. However, it does mean that there could have been patients on the trial who had targetable alterations and that regional differences could have impacted the findings. It would be appropriate for the authors to include known genomic profiling for the patients so that the findings can best be interpreted in the context of molecular alterations.

	Reviewer #1's comments	Author response and changes made
1	In the study protocol and the manuscript, it is mentioned, that the aim was to investigate "similarity" between the study groups. Nevertheless, according to the sample size calculation and the methods used, the study was designed to test for superiority of one treatment option over the other. If the primary goal was to investigate non-inferiority of one treatment as compared to the other or equivalence between the treatment options, this should have been addressed in the planning phase and in the study protocol. Generally, "similarity" does not appear to be a well-defined term.	Thank you for this feedback. This was a one-sided study with a non-inferiority design and a 15% non-inferiority margin. Given the large type I error rate (10%) and the small sample size, we are hesitant to overstate the study results. Despite the one-sided design, two-sided 80% confidence intervals (CIs) are provided, so we are not entirely limited to a one-sided analysis. As a result of the non-inferiority margin, large error rate and small sample size, it was felt that the term 'similarity' was more appropriate. To clarify the study design to the reader, we have added the following text (in red) to the Methods section (page 17): "The planned sample size for the study was 240 (approximately 120 patients in each arm), providing 85% power to detect a 15% difference in ORR between treatment groups (with a 10% one-sided type I error rate) if the true ORR was 45% for both treatment groups. The planned sample size for the study was 240 (approximately 120 patients in each arm), providing 85% power to detect a 15% difference in ORR between treatment groups (with a 10% one-sided type I error rate) if the true ORR was 45% for both treatment groups. 'Similarity' was implied as the ORR for dostarlimab plus chemotherapy not being >15% lower than the ORR for pembrolizumab plus chemotherapy." We have also added further explanation of the reasoning outlined above to the Discussion section (additional text in red) (page 13): "In terms of limitations, this is a Phase II study with a small sample size that was not designed to be powered to statistically confirm superiority. While the study design was based on a non-inferiority trial, the large non-inferiority margin and type I error rate (dictated by consideration of the feasible sample size) precluded a hypothesis of true non-inferiority. As such, the pre-specified hypothesis was that the two treatments were

		'similar'. This limits inferences of differences between the two PD-1 inhibitors."
2	Additionally, in the sample size calculation a one-sided test is mentioned, but I was not able to find the pre-specified direction of the hypotheses to be tested.	Thank you for flagging this point. The pre-specified direction of the hypothesis was dostarlimab non-inferiority. As above, we have added a definition of 'similarity' to the Methods section that indicates the pre-specified direction (page 17): "'Similarity' was implied as the ORR for dostarlimab plus chemotherapy not being >15% lower than the ORR for pembrolizumab plus chemotherapy."
3	In the study protocol and the manuscript, a sample size calculation is presented. For the sample size calculation, a "10% one-sided type I error rate" was used and sample size was determined to allow detection of a difference in ORR of 15 percentage points between the groups with appropriate power. While the chosen significance level is very liberal, I think it is acceptable to use it in a phase II randomized trial. Despite the clear and proper sample size calculation it is mentioned in the discussion that the "study was not formally designed to demonstrate, or statistically powered to test for, superiority". This should be clarified or discussed in more detail.	Thank you for this feedback. As above, we have added further explanation of why this study was not powered to statistically confirm superiority to the Discussion section (additional text in red) (page 13): "In terms of limitations, this is a Phase II study with a small sample size that was not designed to be powered to statistically confirm superiority. While the study design was based on a non-inferiority trial, the large non-inferiority margin and type I error rate (dictated by consideration of the feasible sample size) precluded a hypothesis of true non-inferiority. As such, the pre-specified hypothesis was that the two treatments were 'similar'. This limits inferences of differences between the two PD-1 inhibitors."
4	In the primary analysis a difference in ORR of 9 percentage points was observed. This is smaller than the 15 percentage points considered in the sample size calculation but lead to a statistically significant difference between the groups (no p values are shown, but the 80% confidence interval for the difference in ORR between the groups does not cover the value of zero). For PFS a hazard ratio of 0.70 and difference in median event times of two months were observed. While it is a matter of debate, whether these observed differences can be interpreted as clinically relevant differences between the study groups, I	Thanks for this comment. Due to the large type 1 error rate and the small sample size, we are cautious about overstating the differences shown. While the 80% CI for the difference in ORR between dostarlimab and pembrolizumab does not cross 0, the lower bound of the 95% CI for the same point estimate is below 0% (-2.70–21.33%). The upper bound of the 95% CI for PFS HR is just below 1 (0.98); however, PFS was a secondary endpoint, limiting the conclusions that can be drawn from this observation.

	believe that this should be discussed in more detail. In my opinion, just stating that both groups showed similar results oversimplifies the discussion of the design aspects and the results observed.	Nonetheless, we agree that this study has scientific and clinical novelty as the first large global head-to-head comparison of two PD-1 inhibitors and have added the following short paragraph in red to the discussion section (page 12–13): “There is interest among the scientific and clinical community in the comparability of PD-1 inhibitors and whether they show clinically relevant intra-class differences. Cross-trial comparisons suggest that there may be differences in efficacy and safety¹⁶ and comparison of pharmacokinetic and pharmacodynamic parameters such as binding site, affinity, and half-life have also shown a degree of variability¹⁷. However, PERLA represents the first large global study to compare two PD-1 inhibitors head-to-head in the same indication. While this study was not powered to assess superiority, the two-sided 80% CI for the difference in ORR between dostarlimab and pembrolizumab does not cross 0, suggesting that there may be clinically relevant differences between the two PD-1 inhibitors. As discussed above, future analyses of pharmacodynamic and pharmacokinetic data will provide further insight into these differences and their potential relevance for clinical practice.” References added: 16. Passiglia F, et al. Looking for the best immune-checkpoint inhibitor in pre-treated NSCLC patients: An indirect comparison between nivolumab, pembrolizumab and atezolizumab. Int J Cancer 142, 1277-1284 (2018). 17. Rofi E, et al. Clinical pharmacology of monoclonal antibodies targeting anti-PD-1 axis in urothelial cancers. Crit Rev Oncol Hematol 144, 102812 (2019).
5	Quantification of follow-up times should be reconsidered (see Schemper & Smith, Controlled Clinical Trials, 1996).	Thank you for this comment. We agree that Kaplan–Meier estimate of potential follow-up (KM-PF) is a more suitable method of estimating duration of follow up, in line with the conclusions from Schemper & Smith (1996). We have recalculated the median follow up using this

		methodology and updated the manuscript methods (page 18) and results accordingly (page 5).
6	Number of decimals should be presented more consistently (for example same number of decimals for point estimates and confidence intervals or for means and standard deviations).	Thank you for flagging this inconsistency to us. We have carefully reviewed and ensured that the number of decimal places is aligned between point estimates/means/medians and their corresponding confidence intervals/ranges/standard deviations.
7	For subgroup analyses, test on interactions should be added to investigate heterogeneity of treatment effects.	Thank you for this feedback. Unfortunately, the sample size of subgroups (in particular, the TPS $\geq 50\%$ subgroup, where $n=27$ for both arms) limits the power to detect interactions. We would like to highlight Figure 3, a forest plot of differences in ORR between treatment arms, which illustrates the heterogeneity of treatment effect within the PD-L1 subgroups.
8	Table 2 (ORR by PD-L1 TPS subgroup) is not well formatted and therefore difficult to read.	Thank you for flagging this issue. We have corrected a column width error in the last row of Table 2 (page 28) to ensure that ORRs and 95% CIs align with their respective labels.
9	In Table 3, absolute and relative frequencies of PFS events are shown. While absolute numbers are fine, relative frequencies should be deleted, as these might be misleading due to different lengths of follow-up.	Thanks for this feedback. The percentage of PFS and OS events has been removed from Table 3 (page 30), so that only the raw number of events is presented.
10	Table 4 is not well labelled. Header of the first column is "Median PFS", but also subgroup sizes and hazard ratios are shown.	Thank you for this feedback. We have removed the header from the top of the first column of Table 4 (page 31) and added 'median PFS' to the relevant part of the individual rows.
	Reviewer #2's comments	Author responses and changes made
1	The inclusion of patients based on local testing is to be applauded. Was any central reproducibility testing done?	Thanks for your feedback. Central reproducibility testing was not performed.
2	How do the authors anticipate using this data as it was not designed for superiority study or formal non-inferiority?	These data are intended to support further investigation of dostarlimab as a backbone for novel combinations in NSCLC, by demonstrating that dostarlimab, in combination with chemotherapy, has similar efficacy to the standard of care for first line treatment of metastatic non-squamous NSCLC. This is highlighted in the manuscript conclusion (page 13).
3	Please reformat the tables for readability, the CI are not tracking on-line with the data.	Thank you for flagging this issue. We have corrected a column width error in the last row of Table 2 (page 28) to ensure that ORRs and 95% CIs align with their respective labels.

4	Can the grade 5 event of myelosuppression be more appropriately coded? Is this aplastic anemia? Myelosuppression is not a CTCAE term.	[REDACTED DUE TO PATIENT CONFIDENTIALITY]
	Reviewer #3's comments	Author responses and changes made
1	The authors do not outline the rates of usage of cisplatin vs carboplatin amongst the patients. Although this is unlikely to significantly change the results in any substantive way, given the international nature of the trial, it could give hints as to regional differences in practice patterns or outcomes.	Thank you for highlighting this point. We have added the overall proportion of patients receiving carboplatin and cisplatin to the 'Study Population and Baseline Characteristics' subsection of the Results (page 6). As you will see, the proportion of patients receiving cisplatin was low in both arms (18% in dostarlimab plus chemotherapy arm and 11% in pembrolizumab plus chemotherapy arm). Given these small subgroup sample sizes, more in-depth analysis by region would be difficult to interpret and is therefore not feasible.
2	The authors indicate in the study design that eligible patients were defined as not having "genomic alterations for which an approved targeted therapy was regionally available." This is a practical, real-world consideration that I think is appropriate. However, it does mean that there could have been patients on the trial who had targetable alterations and that regional differences could have impacted the findings. It would be appropriate for the authors to include known genomic profiling for the patients so that the findings can best be interpreted in the context of molecular alterations.	We appreciate the reviewer's commentary with regards to the presence of known genomic aberrations in the patients enrolled in PERLA. This is indeed a very relevant point. However, we would like to clarify that in order to be eligible to enroll in PERLA, patients had to have documented absence of genomic aberrations with approved targeted therapies in their country. We only required testing for exclusionary aberrations and therefore do not have results for all aberrations from all patients. While some sites did use NGS to enroll patients, the majority did not and only tested for relevant aberrations. Hence, any attempt to assess the frequency or impact of such genomic aberrations would be limited.

REVIEWERS' COMMENTS

Reviewer #1 (Remarks to the Author):

Thank you very much for responding to my comments and making changes in the manuscript.

There is just one issue that I think should be addressed:

In my opinion, the wording in your sample size justification suggests a superiority study ("... providing 85% power to detect a 15% difference in ORR between treatment groups ...").

That is also why I thought it was a superiority study.

I would suggest to change the wording using terms commonly used for non-inferiority designs ("non-inferiority margin", etc.).

Reviewer #3 (Remarks to the Author):

In the revised manuscript from Lim et al., the authors report the primary outcomes from a randomized, double-blind, international phase II trial comparing two different anti-PD-1 antibodies in combination with chemotherapy for newly diagnosed patients with metastatic non-squamous NSCLC. This phase II trial represents the first to directly compare two anti-PD-1 antibodies. The revised manuscript is well-written, easy to read, with clear presentation of the data and careful interpretations of the findings. To my mind they have appropriately addressed the concerns raised by the initial reviews. I applaud them for completing this important international trial, which I believe is very timely and of considerable interest to both the lung cancer and broader oncology communities.

	Reviewer #1's comments	Author response and changes made
1	In my opinion, the wording in your sample size justification suggests a superiority study ("... providing 85% power to detect a 15% difference in ORR between treatment groups ..."). That is also why I thought it was a superiority study. I would suggest to change the wording using terms commonly used for non-inferiority designs ("non-inferiority margin", etc.).	Thank you for this feedback. We appreciate the need for clarity on our study design; however, we feel that it would be confusing and potentially misleading to include the term 'non-inferiority margin' as this study was not designed within a rigorous non-inferiority framework. We have further clarified the statistical design the content below (additional content in red, deleted content in strikethrough) in the statistical analysis subsection of the Methods (line 359–63, page 18): "The planned sample size for the study was 240 (approximately 120 patients in each arm), providing 85% power to detect a show that ORR for dostarlimab plus chemotherapy is not more than 15% difference in ORR between treatment groups worse than pembrolizumab plus chemotherapy, (with a at the 10% one-sided type I error rate) level of significance, assuming if the true ORR was 45% for both treatment groups. 'Similarity' was implied as the ORR for dostarlimab plus chemotherapy not being >15% lower than the ORR for pembrolizumab plus chemotherapy."

REVIEWERS' COMMENTS

Reviewer #1 (Remarks to the Author):

I reviewed the adapted version of the manuscript including the updated OS information. I have some concerns with the newest version of the manuscript.

1) Inclusion of the updated OS data makes the manuscript much harder to read.

2) In my opinion, some inconsistencies are introduced, as e.g. for OS and PFS different follow-up times are considered.

3) In the discussion section, it is described that "separation of the OS curves for dostarlimab plus chemotherapy and pembrolizumab plus chemotherapy was evident from 6 months". This is not mentioned in the results section and no corresponding analysis (as e.g. for a time-dependent treatment-effect) was performed. Consequently, I suggest to either extend the analysis and clearly label this analysis and finding as an exploratory post-hoc analysis or to weaken the wording in the discussion section.

Reviewer #3 (Remarks to the Author):

In the further revision of the manuscript from Lim et al., the authors report updated data from a preplanned analysis on the primary outcomes from a randomized, double-blind, international phase II trial comparing two different anti-PD-1 antibodies in combination with chemotherapy for first line treatment of patients with metastatic non-squamous NSCLC. This phase II trial represents the first to directly compare two anti-PD-1 antibodies. The updated data from the recent data cutoff is reassuring of their prior presentation and interpretations of the trial. They should be congratulated for completing this important international trial and in taking the additional effort to update the manuscript prior to its original publication. The work is very timely and of considerable interest to both the lung cancer and broader oncology communities.

Reviewer 1 Comment	Response
1 Inclusion of the updated OS data makes the manuscript much harder to read.	We appreciate your feedback. We feel the most recent data-cut should be reported in the manuscript to tell a more complete story and so the manuscript reports the data with the longest follow-up available. To help orient the reader and clarify which data are included in the main manuscript versus the Supplementary Information, we have added the following paragraph at the beginning of the Results section (new text in blue): LINE 105–115: ‘Two data cuts have been used for the analyses in this manuscript. PERLA primary analyses were conducted using data available as of August 4, 2022. Due to the immature overall survival (OS) and duration of response (DOR) data from the primary analyses, additional analyses were planned for more mature data and better understanding of these endpoints; pre-planned updated analyses were conducted using data available as of July 7, 2023. To provide data on the primary outcome (ORR) and safety using the longest available follow-up, ORR and safety analyses from the July 7, 2023 data cut are included in the main manuscript; results from the August 4, 2022 data cut are available in the Supplementary Information. PFS was a secondary endpoint, and results from the August 4 data cut are presented here in the main text; based on the maturity of data at the time of the primary analysis, PFS was not re-analysed in the updated analysis. OS and DoR analyses using the July 7, 2023 data cut are reported in the main manuscript.’
2 In my opinion, some inconsistencies are introduced, as e.g. for OS and PFS different follow-up times are considered.	Thank you for your comment. While we agree having two follow-up periods make the manuscript more complex, we have included content throughout the manuscript to clearly indicate the follow-up periods and which data-cut is used for each analysis. PFS was a secondary endpoint, and the data were 57% mature (138 events in a total of 243 patients) at the time of the primary analysis. Therefore, as the PFS data were considered mature, no additional analyses of PFS were planned with the new data cut.

3

In the discussion section, it is described that "separation of the OS curves for dostarlimab plus chemotherapy and pembrolizumab plus chemotherapy was evident from 6 months". This is not mentioned in the results section an no corresponding analysis (as e.g. for a time-dependent treatment-effect) was performed. Consequently, I suggest to either extend the analysis and clearly label this analysis and finding as an exploratory post-hoc analysis or to weaken the wording in the discussion section.

Thank you for your feedback. We have not performed time-dependant treatment-effect analyses and so have revised the language in the discussion as follows (new text in blue):

LINE 294–296: *'A pre-planned updated analysis of the July 7, 2023 data cut provided longer survival follow-up and more mature OS data, and demonstrated a numerical trend in median OS favoring dostarlimab plus chemotherapy versus pembrolizumab plus chemotherapy (19.4 months vs 15.9 months, respectively); separation of the OS curves for dostarlimab plus chemotherapy and pembrolizumab plus chemotherapy appears to start ~~was~~ evident from 6 months as per the Kaplan-Meier curves.*